# Learning to Optimize via Information-Directed Sampling

**Daniel Russo**
Stanford University
Stanford, CA 94305
djrusso@stanford.edu

**Benjamin Van Roy**
Stanford University
Stanford, CA 94305
bvr@stanford.edu

## Abstract

We propose *information-directed sampling* – a new algorithm for online optimization problems in which a decision-maker must balance between exploration and exploitation while learning from partial feedback. Each action is sampled in a manner that minimizes the ratio between the square of expected single-period regret and a measure of information gain: the mutual information between the optimal action and the next observation.

We establish an expected regret bound for information-directed sampling that applies across a very general class of models and scales with the entropy of the optimal action distribution. For the widely studied Bernoulli and linear bandit models, we demonstrate simulation performance surpassing popular approaches, including upper confidence bound algorithms, Thompson sampling, and knowledge gradient. Further, we present simple analytic examples illustrating that information-directed sampling can dramatically outperform upper confidence bound algorithms and Thompson sampling due to the way it measures information gain.

## 1 Introduction

There has been significant recent interest in extending multi-armed bandit techniques to address problems with more complex *information structures*, in which sampling one action can inform the decision-maker's assessment of other actions. Effective algorithms must take advantage of the information structure to learn more efficiently. Recent work has extended popular algorithms for the classical multi-armed bandit problem, such as *upper confidence bound* (UCB) algorithms and *Thompson sampling*, to address such contexts.

For some cases, such as classical and linear bandit problems, strong performance guarantees have been established for UCB algorithms (e.g. [4, 8, 9, 13, 21, 23, 29]) and Thompson sampling (e.g. [1, 15, 19, 24]). However, as we will demonstrate through simple analytic examples, these algorithms can perform very poorly when faced with more complex information structures. The shortcoming lies in the fact that these algorithms do not adequately assess the information gain from selecting an action.

In this paper, we propose a new algorithm – *information-directed sampling* (IDS) – that preserves numerous guarantees of Thompson sampling for problems with simple information structures while offering strong performance in the face of more complex problems that daunt alternatives like Thompson sampling or UCB algorithms. IDS quantifies the amount learned by selecting an action through an information theoretic measure: the mutual information between the true optimal action and the next observation. Each action is sampled in a manner that minimizes the ratio between squared expected single-period regret and this measure of information gain.

As we will show through simple analytic examples, the way in which IDS assesses information gain allows it to dramatically outperform UCB algorithms and Thompson sampling. Further, we establish

an expected regret bound for IDS that applies across a very general class of models and scales with the entropy of the optimal action distribution. We then specialize this bound to several widely studied problem classes. Finally, we benchmark the performance of IDS through simulations of the widely studied Bernoulli and linear bandit problems, for which UCB algorithms and Thompson sampling are known to be very effective. We find that even in these settings, IDS outperforms UCB algorithms, Thompson sampling, and knowledge gradient.

IDS solves a single-period optimization problem as a proxy to an intractable multi-period problem. Solution of this single-period problem can itself be computationally demanding, especially in cases where the number of actions is enormous or mutual information is difficult to evaluate. To carry out computational experiments, we develop numerical methods for particular classes of online optimization problems. More broadly, we feel this work provides a compelling proof of concept and hope that our development and analysis of IDS facilitate the future design of efficient algorithms that capture its benefits.

**Related literature.** Two other papers [17, 30] have used the mutual information between the optimal action and the next observation to guide action selection. Both focus on the optimization of expensive-to-evaluate, black-box functions. Each proposes sampling points so as to maximize the mutual information between the algorithm's next observation and the true optimizer. Several features distinguish our work. First, these papers focus on pure exploration problems: the objective is simply to learn about the optimum – not to attain high cumulative reward. Second, and more importantly, they focus only on problems with Gaussian process priors and continuous action spaces. For such problems, simpler approaches like UCB algorithms, Probability of Improvement, and Expected Improvement are already extremely effective (See [6]). By contrast, a major motivation of our work is that a richer information measure is needed in order to address problems with more complicated information structures. Finally, we provide a variety of general theoretical guarantees for IDS, whereas Villemonteix et al. [30] and Hennig and Schuler [17] propose their algorithms only as heuristics. The full-length version of this paper [26] shows our theoretical guarantees extend to pure exploration problems.

The knowledge gradient (KG) algorithm uses a different measure of information to guide action selection: the algorithm computes the impact of a single observation on the quality of the decision made by a *greedy* algorithm, which simply selects the action with highest posterior expected reward. This measure has been thoroughly studied (see e.g. [22, 27]). KG seems natural since it explicitly seeks information that improves decision quality. Computational studies suggest that for problems with Gaussian priors, Gaussian rewards, and relatively short time horizons, KG performs very well. However, even in some simple settings, KG may not converge to optimality. In fact, it may select a suboptimal action in *every* period, even as the time horizon tends to infinity.

Our work also connects to a much larger literature on Bayesian experimental design (see [10] for a review). Recent work has demonstrated the effectiveness of *greedy* or *myopic* policies that always maximize the information gain the next sample. Jedynak et al. [18] consider problem settings in which this greedy policy is optimal. Another recent line of work [14] shows that information gain based objectives sometimes satisfy a decreasing returns property known as adaptive sub-modularity, implying the greedy policy is competitive with the optimal policy. Our algorithm also only considers only the information gain due to the *next sample*, even though the goal is to acquire information over many periods. Our results establish that the manner in which IDS encourages information gain leads to an effective algorithm, even for the different objective of maximizing cumulative reward.

## 2 Problem formulation

We consider a general probabilistic, or Bayesian, formulation in which uncertain quantities are modeled as random variables. The decision–maker sequentially chooses actions $(A_t)_{t\in\mathbb{N}}$ from the finite action set $\mathcal{A}$ and observes the corresponding outcomes $(Y_t(A_t))_{t\in\mathbb{N}}$. There is a random outcome $Y_t(a) \in \mathcal{Y}$ associated with each $a \in \mathcal{A}$ and time $t \in \mathbb{N}$. Let $Y_t \equiv (Y_t(a))_{a\in\mathcal{A}}$ be the vector of outcomes at time $t \in \mathbb{N}$. The "true outcome distribution" $p^*$ is a distribution over $\mathcal{Y}^{|\mathcal{A}|}$ that is itself randomly drawn from the family of distributions $\mathcal{P}$. We assume that, conditioned on $p^*$, $(Y_t)_{t\in\mathbb{N}}$ is an iid sequence with each element $Y_t$ distributed according to $p^*$. Let $p_a^*$ be the marginal distribution corresponding to $Y_t(a)$.

The agent associates a reward $R(y)$ with each outcome $y \in \mathcal{Y}$, where the reward function $R : \mathcal{Y} \to \mathbb{R}$ is fixed and known. We assume $R(\overline{y}) - R(\underline{y}) \le 1$ for any $\overline{y}, \underline{y} \in \mathcal{Y}$. Uncertainty about $p^*$ induces uncertainty about the true optimal action, which we denote by $A^* \in \arg\max\limits_{a \in \mathcal{A}} \mathbb{E}\limits_{y \sim p_a^*} [R(y)]$. The $T$ period *regret* is the random variable,

$$\text{Regret}(T) := \sum_{t=1}^{T} [R(Y_t(A^*)) - R(Y_t(A_t))], \tag{1}$$

which measures the cumulative difference between the reward earned by an algorithm that always chooses the optimal action, and actual accumulated reward up to time $T$. In this paper we study expected regret $\mathbb{E}[\text{Regret}(T)]$ where the expectation is taken over the randomness in the actions $A_t$ and the outcomes $Y_t$, and over the prior distribution over $p^*$. This measure of performance is sometimes called *Bayesian regret* or *Bayes risk*.

**Randomized policies.** We define all random variables with respect to a probability space $(\Omega, \mathcal{F}, \mathbb{P})$. Fix the filtration $(\mathcal{F}_t)_{t \in \mathbb{N}}$ where $\mathcal{F}_{t-1} \subset \mathcal{F}$ is the sigma–algebra generated by the history of observations $(A_1, Y_1(A_1), ..., A_{t-1}, Y_{t-1}(A_{t-1}))$. Actions are chosen based on the history of past observations, and possibly some external source of randomness[1]. It's useful to think of the actions as being chosen by a *randomized policy* $\pi$, which is an $\mathcal{F}_t$–predictable sequence $(\pi_t)_{t \in \mathbb{N}}$. An action is chosen at time $t$ by randomizing according to $\pi_t(\cdot) = \mathbb{P}(A_t = \cdot | \mathcal{F}_{t-1})$, which specifies a probability distribution over $\mathcal{A}$. We denote the set of probability distributions over $\mathcal{A}$ by $\mathcal{D}(\mathcal{A})$. We explicitly display the dependence of regret on the policy $\pi$, letting $\mathbb{E}[\text{Regret}(T, \pi)]$ denote the expected value of (1) when the actions $(A_1, .., A_T)$ are chosen according to $\pi$.

**Further notation.** We set $\alpha_t(a) = \mathbb{P}(A^* = a | \mathcal{F}_{t-1})$ to be the posterior distribution of $A^*$. For a probability distribution $P$ over a finite set $\mathcal{X}$, the *Shannon entropy* of $P$ is defined as $H(P) = -\sum_{x \in \mathcal{X}} P(x) \log(P(x))$. For two probability measures $P$ and $Q$ over a common measurable space, if $P$ is absolutely continuous with respect to $Q$, the *Kullback-Leibler divergence* between $P$ and $Q$ is

$$D_{\text{KL}}(P || Q) = \int_{\mathcal{Y}} \log\left(\frac{dP}{dQ}\right) dP \tag{2}$$

where $\frac{dP}{dQ}$ is the Radon–Nikodym derivative of $P$ with respect to $Q$. The *mutual information* under the posterior distribution between random variables $X_1 : \Omega \to \mathcal{X}_1$, and $X_2 : \Omega \to \mathcal{X}_2$, denoted by

$$I_t(X_1; X_2) := D_{\text{KL}}\left(\mathbb{P}((X_1, X_2) \in \cdot | \mathcal{F}_{t-1}) \ || \ \mathbb{P}(X_1 \in \cdot | \mathcal{F}_{t-1}) \mathbb{P}(X_2 \in \cdot | \mathcal{F}_{t-1})\right), \tag{3}$$

is the Kullback-Leibler divergence between the joint posterior distribution of $X_1$ and $X_2$ and the product of the marginal distributions. Note that $I_t(X_1; X_2)$ is a random variable because of its dependence on the conditional probability measure $\mathbb{P}(\cdot | \mathcal{F}_{t-1})$.

To simplify notation, we define the *information gain* from an action $a$ to be $g_t(a) := I_t(A^*; Y_t(a))$. As shown for example in Lemma 5.5.6 of Gray [16], this is equal to the expected reduction in entropy of the posterior distribution of $A^*$ due to observing $Y_t(a)$:

$$g_t(a) = \mathbb{E}[H(\alpha_t) - H(\alpha_{t+1}) | \mathcal{F}_{t-1}, A_t = a], \tag{4}$$

which plays a crucial role in our results. Let $\Delta_t(a) := \mathbb{E}[R_t(Y_t(A^*)) - R(Y_t(a)) | \mathcal{F}_{t-1}]$ denote the expected instantaneous regret of action $a$ at time $t$. We overload the notation $g_t(\cdot)$ and $\Delta_t(\cdot)$. For $\pi \in \mathcal{D}(\mathcal{A})$, define $g_t(\pi) = \sum_{a \in \mathcal{A}} \pi(a) g_t(a)$ and $\Delta_t(\pi) = \sum_{a \in \mathcal{A}} \pi(a) \Delta_t(a)$.

## 3   Information-directed sampling

IDS explicitly balances between having low expected regret in the current period and acquiring new information about which action is optimal. It does this by maximizing over all action sampling distributions $\pi \in \mathcal{D}(\mathcal{A})$ the ratio between the square of expected regret $\Delta_t(\pi)^2$ and information

gain $g_t(\pi)$ about the optimal action $A^*$. In particular, the policy $\pi^{\mathrm{IDS}} = \left( \pi_1^{\mathrm{IDS}}, \pi_2^{\mathrm{IDS}}, ... \right)$ is defined by:

$$\pi_t^{\mathrm{IDS}} \in \arg\min_{\pi \in \mathcal{D}(\mathcal{A})} \left\{ \Psi_t(\pi) := \frac{\Delta_t(\pi)^2}{g_t(\pi)} \right\}. \tag{5}$$

We call $\Psi_t(\pi)$ the *information ratio* of a sampling distribution $\pi$ and $\Psi_t^* = \min_\pi \Psi_t(\pi) = \Psi_t(\pi_t^{\mathrm{IDS}})$ the *minimal information ratio*. Each roughly measures the "cost" per bit of information acquired.

**Optimization problem.** Suppose that there are $K = |\mathcal{A}|$ actions, and that the posterior expected regret and information gain are stored in the vectors $\Delta \in \mathbb{R}_+^K$ and $g \in \mathbb{R}_+^K$. Assume $g \neq 0$, so that the optimal action is not known with certainty. The optimization problem (5) can be written as

$$\text{minimize} \quad \Psi(\pi) := \left( \pi^T \Delta \right)^2 / \pi^T g \quad \text{subject to} \quad \pi^T e = 1, \ \pi \geq 0. \tag{6}$$

The following result shows this is a convex optimization problem, and surprisingly, has an optimal solution with only two non-zero components. Therefore, while IDS is a randomized policy, it randomizes over at most two actions. Algorithm 1, presented in the supplementary material, solves (6) by looping over all pairs of actions, and solving a one dimensional convex optimization problem.

**Proposition 1.** *The function* $\Psi : \pi \mapsto \left( \pi^T \Delta \right)^2 / \pi^T g$ *is convex on* $\left\{ \pi \in \mathbb{R}^K | \pi^T g > 0 \right\}$. *Moreover, there is an optimal solution* $\pi^*$ *to* (6) *with* $|\{ i : \pi_i^* > 0 \}| \leq 2$.

## 4   Regret bounds

This section establishes regret bounds for IDS that scale with the entropy of the optimal action distribution. The next proposition shows that bounds on a policy's information ratio imply bounds on expected regret. We then provide several bounds on the information ratio of IDS.

**Proposition 2.** *Fix a deterministic* $\lambda \in \mathbb{R}$ *and a policy* $\pi = (\pi_1, \pi_2, ...)$ *such that* $\Psi_t(\pi_t) \leq \lambda$ *almost surely for each* $t \in \{1, .., T\}$. *Then,* $\mathbb{E}\left[ \text{Regret}(\pi, T) \right] \leq \sqrt{\lambda H(\alpha_1) T}$.

**Bounds on the information ratio.** We establish upper bounds on the minimal information ratio $\Psi_t^* = \Psi_t^*(\pi_t^{\mathrm{IDS}})$ in several important settings. These bound show that, in any period, the algorithm's expected regret can only be large if it's expected to acquire a lot of information about which action is optimal. It effectively balances between exploration and exploitation in *every* period.

The proofs of these bounds essentially follow from a very recent analysis of Thompson sampling, and the implied regret bounds are the same as those established for Thompson sampling. In particular, since $\Psi_t^* \leq \Psi_t(\pi^{\mathrm{TS}})$ where $\pi^{\mathrm{TS}}$ is the Thompson sampling policy, it is enough to bound $\Psi_t(\pi^{\mathrm{TS}})$. Several such bounds were provided by Russo and Van Roy [25].[2] While the analysis is similar in the cases considered here, IDS outperforms Thompson sampling in simulation, and, as we will highlight in the next section, is sometimes provably much more informationally efficient.

We briefly describe each of these bounds below and then provide a more complete discussion for linear bandit problems. For each of the other cases, more formal propositions, their proofs, and a discussion of lower bounds can be found in the supplementary material or the full version of this paper [26].

**Finite action space:** With no additional assumption, we show $\Psi_t^* \leq |\mathcal{A}|/2$.

**Linear bandit:** Each action is associated with a $d$ dimensional feature vector, and the mean reward generated by an action is the inner product between its known feature vector and some unknown parameter vector. We show $\Psi_t^* \leq d/2$.

**Full information:** Upon choosing an action, the agent observes the reward she would have received had she chosen any other action. We show $\Psi_t^* \leq 1/2$.

**Combinatorial action sets:** At time $t$, project $i \in \{1, .., d\}$ yields a random reward $\theta_{t,i}$, and the reward from selecting a subset of projects $a \in \mathcal{A} \subset \{a' \subset \{0, 1, ..., d\} : |a'| \leq m\}$ is $m^{-1} \sum_{i \in \mathcal{A}} \theta_{t,i}$. The outcome of each selected project $(\theta_{t,i} : i \in a)$ is observed, which is sometimes called "semi–bandit" feedback [3]. We show $\Psi_t^* \leq d/2m^2$.

**Linear optimization under bandit feedback.** The stochastic linear bandit problem has been widely studied (e.g. [13, 23]) and is one of the most important examples of a multi-armed bandit problem with "correlated arms." In this setting, each action is associated with a finite dimensional feature vector, and the mean reward generated by an action is the inner product between its known feature vector and some unknown parameter vector. The next result bounds $\Psi_t^*$ for such problems.

**Proposition 3.** *If $\mathcal{A} \subset \mathbb{R}^d$ and for each $p \in \mathcal{P}$ there exists $\theta_p \in \mathbb{R}^d$ such that for all $a \in \mathcal{A}$ $\mathop{\mathbb{E}}_{y \sim p_a}[R(y)] = a^T \theta_p$, then for all $t \in \mathbb{N}$, $\Psi_t^* \leq d/2$ almost surely.*

This result shows that $\mathbb{E}\left[\mathrm{Regret}(T, \pi^{\mathrm{IDS}})\right] \leq \sqrt{\frac{1}{2}H(\alpha_1)dT} \leq \sqrt{\frac{1}{2}\log(|\mathcal{A}|)dT}$ for linear bandit problems. Dani et al. [12] show this bound is order optimal, in the sense that for any time horizon $T$ and dimension $d$ if the actions set is $\mathcal{A} = \{0, 1\}^d$, there exists a prior distribution over $p^*$ such that $\inf_\pi \mathbb{E}\left[\mathrm{Regret}(T, \pi)\right] \geq c_0 \sqrt{\log(|\mathcal{A}|)dT}$ where $c_0$ is a constant the is independent of $d$ and $T$. The bound here improves upon this worst case bound since $H(\alpha_1)$ can be much smaller than $\log(|\mathcal{A}|)$.

## 5 Beyond UCB and Thompson sampling

Upper confidence bound algorithms (UCB) and Thompson sampling are two of the most popular approaches to balancing between exploration and exploitation. In some cases, these algorithms are empirically effective, and have strong theoretical guarantees. But we will show that, because they don't quantify the information provided by sampling actions, they can be grossly suboptimal in other cases. We demonstrate this through two examples - each designed to be simple and transparent. To set the stage for our discussion, we now introduce UCB algorithms and Thompson sampling.

**Thompson sampling.** The Thompson sampling algorithm simply samples actions according to the posterior probability they are optimal. In particular, actions are chosen randomly at time $t$ according to the sampling distribution $\pi_t^{\mathrm{TS}} = \alpha_t$. By definition, this means that for each $a \in \mathcal{A}$, $\mathbb{P}(A_t = a | \mathcal{F}_{t-1}) = \mathbb{P}(A^* = a | \mathcal{F}_{t-1}) = \alpha_t(a)$. This algorithm is sometimes called *probability matching* because the action selection distribution is *matched* to the posterior distribution of the optimal action. Note that Thompson sampling draws actions only from the support of the posterior distribution of $A^*$. That is, it never selects an action $a$ if $\mathbb{P}(A^* = a) = 0$. Put differently, this implies that it only selects actions that are optimal under some $p \in \mathcal{P}$.

**UCB algorithms.** UCB algorithms select actions through two steps. First, for each action $a \in \mathcal{A}$ an upper confidence bound $B_t(a)$ is constructed. Then, an action $A_t \in \arg\max_{a \in \mathcal{A}} B_t(a)$ with maximal upper confidence bound is chosen. Roughly, $B_t(a)$ represents the greatest mean reward value that is statistically plausible. In particular, $B_t(a)$ is typically constructed so that $B_t(a) \to \mathop{\mathbb{E}}_{y \sim p_a^*}[R(y)]$ as data about action $a$ accumulates, but with high probability $\mathop{\mathbb{E}}_{y \sim p_a^*}[R(y)] \leq B_t(a)$.

Like Thompson sampling, many UCB algorithms only select actions that are optimal under some $p \in \mathcal{P}$. Consider an algorithm that constructs at each time $t$ a confidence set $\mathcal{P}_t \subset \mathcal{P}$ containing the set of distributions that are statistically plausible given observed data. (e.g. [13]). Upper confidence bounds are then set to be the highest expected reward attainable under one of the plausible distributions:

$$B_t(a) = \max_{p \in \mathcal{P}} \mathop{\mathbb{E}}_{y \sim p_a}[R(y)].$$

Any action $A_t \in \arg\max_a B_t(a)$ must be optimal under one of the outcome distributions $p \in \mathcal{P}_t$. An alternative method involves choosing $B_t(a)$ to be a particular quantile of the posterior distribution of the action's mean reward under $p^*$ [20]. In each of the examples we construct, such an algorithm chooses actions from the support of $A^*$ unless the quantiles are so low that $\max_{a \in \mathcal{A}} B_t(a) < \mathbb{E}[R(Y_t(A^*))]$.

### 5.1 Example: sparse linear bandits

Consider a linear bandit problem where $\mathcal{A} \subset \mathbb{R}^d$ and the reward from an action $a \in \mathcal{A}$ is $a^T \theta^*$. The true parameter $\theta^*$ is known to be drawn uniformly at random from the set of 1–sparse vectors $\Theta = \{\theta \in \{0, 1\}^d : \|\theta\|_0 = 1\}$. For simplicity, assume $d = 2^m$ for some $m \in \mathbb{N}$. The action set is taken to be the set of vectors in $\{0, 1\}^d$ normalized to be a unit vector in the $L^1$ norm: $\mathcal{A} =$

$\left\{ \frac{x}{\|x\|_1} : x \in \{0,1\}^d, x \neq 0 \right\}$. We will show that the expected number of time steps for Thompson sampling (or a UCB algorithm) to identify the optimal action grows linearly with $d$, whereas IDS requires only $\log_2(d)$ time steps.

When an action $a$ is selected and $y = a^T \theta^* \in \{0, 1/\|a\|_0\}$ is observed, each $\theta \in \Theta$ with $a^T \theta \neq y$ is ruled out. Let $\Theta_t$ denote the parameters in $\Theta$ that are consistent with the observations up to time $t$ and let $\mathcal{I}_t = \{i \in \{1, ..., d\} : \theta_i = 1, \theta \in \Theta_t\}$ be the set of possible positive components.

For this problem, $A^* = \theta^*$. That is, if $\theta^*$ were known, the optimal action would be to choose the action $\theta^*$. Thompson sampling and UCB algorithms only choose actions from the support of $A^*$ and therefore will only sample actions $a \in \mathcal{A}$ that have only a single positive component. Unless that is also the positive component of $\theta^*$, the algorithm will observe a reward of zero and rule out only one possible value for $\theta^*$. The algorithm may require $d$ samples to identify the optimal action.

Consider an application of IDS to this problem. It essentially performs binary search: it selects $a \in \mathcal{A}$ with $a_i > 0$ for half of the components $i \in \mathcal{I}_t$ and $a_i = 0$ for the other half as well as for any $i \notin \mathcal{I}_t$. After just $\log_2(d)$ time steps the true support of $\theta^*$ is identified.

To see why this is the case, first note that all parameters in $\Theta_t$ are equally likely and hence the expected reward of an action $a$ is $\frac{1}{|\mathcal{I}_t|} \sum_{i \in I_t} a_i$. Since $a_i \geq 0$ and $\sum_i a_i = 1$ for each $a \in \mathcal{A}$, every action whose positive components are in $I_t$ yields the highest possible expected reward of $1/|I_t|$. Therefore, binary search minimizes expected regret in period $t$ for this problem. At the same time, binary search is assured to rule out half of the parameter vectors in $\Theta_t$ at each time $t$. This is the largest possible expected reduction, and also leads to the largest possible information gain about $A*$. Since binary search both minimizes expected regret in period $t$ and uniquely maximizes expected information gain in period $t$, it is the sampling strategy followed by IDS.

## 5.2 Example: recommending products to a customer of unknown type

Consider the problem of repeatedly recommending an assortment of products to a customer. The customer has unknown type $c^* \in C$ where $|C| = n$. Each product is geared toward customers of a particular type, and the assortment $a \in \mathcal{A} = C^m$ of $m$ products offered is characterized by the vector of product types $a = (c_1, .., c_m)$. We model customer responses through a random utility model in which customers are apriori more likely to derive high value from a product geared toward their type. When offered an assortment of products $a$, the customer associates with the $i$th product utility $U_{ci}^{(t)}(a) = \beta \mathbf{1}_{\{a_i = c\}} + W_{ci}^t$, where $W_{ci}^t$ follows an extreme–value distribution and $\beta \in \mathbb{R}$ is a known constant. This is a standard multinomial logit discrete choice model. The probability a customer of type $c$ chooses product $i$ is given by $\exp\{\beta \mathbf{1}_{\{a_i = c\}}\} / \sum_{j=1}^m \exp\{\beta \mathbf{1}_{\{a_j = c\}}\}$. When an assortment $a$ is offered at time $t$, the customer makes a choice $I_t = \arg\max_i U_{ci}^{(t)}(a)$ and leaves a review $U_{cI_t}^{(t)}(a)$ indicating the utility derived from the product, both of which are observed by the recommendation system. The system's reward is the normalized utility of the customer $(\frac{1}{\beta}) U_{cI_t}^{(t)}(a)$.

If the type $c^*$ of the customer were known, then the optimal recommendation would be $A^* = (c^*, c^*, ..., c^*)$, which consists only of products targeted at the customer's type. Therefore, both Thompson sampling and UCB algorithms would only offer assortments consisting of a single type of product. Because of this, each type of algorithm requires order $n$ samples to learn the customer's true type. IDS will instead offer a *diverse* assortment of products to the customer, allowing it to learn much more quickly.

To make the presentation more transparent, suppose that $c^*$ is drawn uniformly at random from $\mathcal{C}$ and consider the behavior of each type of algorithm in the limiting case where $\beta \to \infty$. In this regime, the probability a customer chooses a product of type $c^*$ if it available tends to 1, and the review $U_{cI_t}^{(t)}(a)$ tends to $\mathbf{1}\{a_{I_t} = c^*\}$, an indicator for whether the chosen product had type $c^*$. The initial assortment offered by IDS will consist of $m$ different and previously untested product types. Such an assortment maximizes both the algorithm's expected reward in the next period and the algorithm's information gain, since it has the highest probability of containing a product of type $c^*$. The customer's response almost perfectly indicates whether one of those items was of type $c^*$. The algorithm continues offering assortments containing $m$ unique, untested, product types until a

review near $U_{cI_t}^{(t)}(a) \approx 1$ is received. With extremely high probability, this takes at most $\lceil n/m \rceil$ time periods. By diversifying the $m$ products in the assortment, the algorithm learns $m$ times faster.

# 6 Computational experiments

Section 5 showed that, for some complicated information structures, popular approaches like UCB algorithms and Thompson sampling are provably outperformed by IDS. Our computational experiments focus instead on simpler settings where these algorithms are extremely effective. We find that even for these widely studied settings, IDS displays performance exceeding state of the art. For each experiment, the algorithm used to implement IDS is presented in Appendix C.

**Mean-based IDS.** Some of our numerical experiments use an approximate form of IDS that is suitable for some problems with bandit feedback, satisfies our regret bounds for such problems, and can sometimes facilitate design of more efficient numerical methods. More details can be found in the appendix, or in the full version of this paper [26].

**Beta-Bernoulli experiment.** Our first experiment involves a multi-armed bandit problem with independent arms. The action $a_i \in \{a_1, ..., a_K\}$ yields in each time period a reward that is 1 with probability $\theta_i$ and 0 otherwise. The $\theta_i$ are drawn independently from $\mathrm{Beta}(1, 1)$, which is the uniform distribution. Figure 1a presents the results of 1000 independent trials of an experiment with 10 arms and a time horizon of 1000. We compare IDS to six other algorithms, and find that it has the lowest average regret of 18.16. Our results indicate that the the variation of IDS $\pi^{\mathrm{IDS_{ME}}}$ presented in Section 6 has extremely similar performance to standard IDS for this problem.

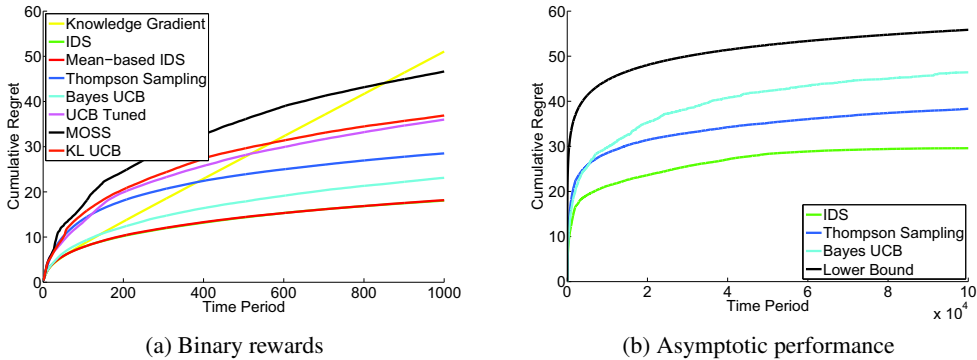

(a) Binary rewards          (b) Asymptotic performance

In this experiment, the famous UCB1 algorithm of Auer et al. [4] had average regret 131.3, which is dramatically larger than that of IDS. For this reason UCB1 is omitted from Figure 1a. The confidence bounds of UCB1 are constructed to facilitate theoretical analysis. For practical performance Auer et al. [4] proposed using a heuristic algorithm called UCB-Tuned. The MOSS algorithm of Audibert and Bubeck [2] is similar to UCB1 and UCB–Tuned, but uses slightly different confidence bounds. It is known to satisfy regret bounds for this problem that are minimax optimal up to a constant factor.

In previous numerical experiments [11, 19, 20, 28], Thompson sampling and Bayes UCB exhibited state-of-the-art performance for this problem. Unsurprisingly, they are the closest competitors to IDS. The Bayes UCB algorithm, studied in Kaufmann et al. [20], uses upper confidence bounds at time step $t$ that are the $1 - \frac{1}{t}$ quantile of the posterior distribution of each action[3].

The knowledge gradient (KG) policy of Ryzhov et al. [27], uses the one–step value of information to incentivize exploration. However, for this problem, KG does not explore sufficiently to identify the optimal arm in this problem, and therefore its expected regret grows linearly with time. It should be noted that KG is particularly poorly suited to problems with discrete observations and long time horizons. It can perform very well in other types of experiments.

**Asymptotic optimality.** That IDS outperforms Bayes UCB and Thompson sampling in our last experiment is is particularly surprising, as each of these algorithms is known, in a sense we will

soon formalize, to be asymptotically optimal for these problems. We now present simulation results over a much longer time horizon that suggest IDS scales in the same asymptotically optimal way.

The seminal work of Lai and Robbins [21] provides the following asymptotic *frequentist* lower bound on regret of any policy $\pi$. When applied with an independent uniform prior over $\theta$, both Bayes UCB and Thompson sampling are known to attain this frequentist lower bound [19, 20]:

$$\liminf_{T \to \infty} \frac{\mathbb{E}\left[\text{Regret}(T, \pi)|\theta\right]}{\log T} \geq \sum_{a \neq A^*} \frac{(\theta_{A^*} - \theta_a)}{D_{\text{KL}}(\theta_{A^*} \,||\, \theta_a)} := c(\theta)$$

Our next numerical experiment fixes a problem with three actions and with $\theta = (.3, .2, .1)$. We compare algorithms over a 10,000 time periods. Due to the computational expense of this experiment, we only ran 200 independent trials. Each algorithm uses a uniform prior over $\theta$. Our results, along with the asymptotic lower bound of $c(\theta)\log(T)$, are presented in Figure 1b.

**Linear bandit problems.** Our final numerical experiment treats a linear bandit problem. Each action $a \in \mathbb{R}^5$ is defined by a 5 dimensional feature vector. The reward of action $a$ at time $t$ is $a^T\theta + \epsilon_t$ where $\theta \sim N(0, 10I)$ is drawn from a multivariate Gaussian prior distribution, and $\epsilon_t \sim N(0, 1)$ is independent Gaussian noise. In each period, only the reward of the selected action is observed. In our experiment, the action set $\mathcal{A}$ contains 30 actions, each with features drawn uniformly at random from $[-1/\sqrt{5}, 1/\sqrt{5}]$. The results displayed in Figure 1 are averaged over 1000 independent trials.

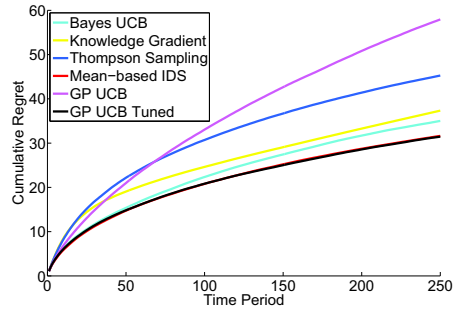

Figure 1: Regret in linear–Gaussian model.

We compare the regret of five algorithms. Three of these - GP-UCB, Thompson sampling , and IDS - satisfy strong regret bounds for this problem[4]. Both GP-UCB and Thompson sampling are significantly outperformed by IDS. Bayes UCB [20] and a version of GP-UCB that was tuned to minimize its average regret, are each competitive with IDS. These algorithms are heuristics, in the sense that their confidence bounds differ significantly from those of linear UCB algorithms known to satisfy theoretical guarantees.

## 7 Conclusion

This paper has proposed information-directed sampling – a new algorithm for balancing between exploration and exploitation. We establish a general regret bound for the algorithm, and specialize this bound to several widely studied classes of online optimization problems. We show the way in which IDS assesses information gain allows it to dramatically outperform UCB algorithms and Thompson sampling in some settings. Finally, for two simple and widely studied classes of multi-armed bandit problems we demonstrate state of art performance in simulation experiments. In these ways, we feel this work provides a compelling proof of concept.

Many important open questions remain, however. IDS solves a single-period optimization problem as a proxy to an intractable multi-period problem. Solution of this single-period problem can itself be computationally demanding, especially in cases where the number of actions is enormous or mutual information is difficult to evaluate. An important direction for future research concerns the development of computationally elegant procedures to implement IDS in important cases. Even when the algorithm cannot be directly implemented, however, one may hope to develop simple algorithms that capture its main benefits. Proposition 2 shows that any algorithm with small information ratio satisfies strong regret bounds. Thompson sampling is a very tractable algorithm that, we conjecture, sometimes has nearly minimal information ratio. Perhaps simple schemes with small information ratio could be developed for other important problem classes, like the sparse linear bandit problem.

## Footnotes

[1]Formally, $A_t$ is measurable with respect to the sigma–algebra generated by $(\mathcal{F}_{t-1}, \xi_t)$ where $(\epsilon_t)_{t \in \mathbb{N}}$ are random variables representing this external source of randomness, and are jointly independent of $p^*$ and $(Y_t)_{t \in \mathbb{N}}$

[2] $\Psi_t(\pi^{\mathrm{TS}})$ is exactly equal to the term $\Gamma_t^2$ that is bounded in [25].

[3]Their theoretical guarantees require choosing a somewhat higher quantile, but the authors suggest choosing this quantile, and use it in their own numerical experiments.

[4]Regret analysis of GP-UCB can be found in [29] and for Thompson sampling can be found in [1, 24, 25]

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
