[Supplementary Material]

# A Mean-based IDS

Here we introduce an approximate form of IDS that is suitable for some problems with bandit feedback, satisfies our regret bounds for such problems, and can sometimes facilitate design of more efficient numerical methods. We will derive this policy by investigating the structure of the mutual information $g_t(a) = I_t(A^*, Y_t(a))$, and considering a modified information measure.

Let $p_{t,a} = \mathbb{P}(Y_t(a) \in \cdot | \mathcal{F}_{t-1})$ denote the posterior predictive distribution at an action $a$, and let $p_{t,a}(\cdot | a^*) = \mathbb{P}(Y_t(a) \in \cdot | \mathcal{F}_{t-1}, A^* = a^*)$ denote the posterior predictive distribution conditional on the event that $a^*$ is the optimal action. Crucial to our results is the following fact, which is a consequence of standard properties of mutual information[5]. It shows, the mutual information between $A^*$ and $Y_t(a)$ is the expected KL divergence between the posterior predictive distribution $p_{t,a}$ and the predictive distribution conditioned on the identity of the optimal action $p_{t,a}(\cdot | a^*)$:

$$g_t(a) = \mathop{\mathbb{E}}_{a^* \sim \alpha_t} \left[ D_{\mathrm{KL}} \left( p_{t,a}(\cdot | a^*) \,||\, p_{t,a} \right) \right]. \tag{7}$$

Our analysis in the full version of this paper [26] provides theoretical guarantees for an algorithm that replaces the Kullback-Leibler divergence in (7) with a simpler measure of divergence: the squared divergence "in mean". Define

$$g_t^{\mathrm{ME}}(a) = \mathop{\mathbb{E}}_{a^* \sim \alpha_t} \left[ D_{\mathrm{ME}} \left( p_{t,a}(\cdot | a^*) \,||\, p_{t,a} \right)^2 \right], \text{ and } D_{\mathrm{ME}}(P \,||\, Q) := \mathop{\mathbb{E}}_{y \sim P} [R(y)] - \mathop{\mathbb{E}}_{y \sim Q} [R(y)].$$

We introduce the policy $\pi^{\mathrm{IDS_{ME}}} = \left( \pi_1^{\mathrm{IDS_{ME}}}, \pi_2^{\mathrm{IDS_{ME}}}, ... \right)$ where $\pi_t^{\mathrm{IDS_{ME}}} \in \operatorname*{arg\,min}_{\pi \in \mathcal{D}(\mathcal{A})} \frac{\Delta_t(\pi)^2}{g_t^{\mathrm{ME}}(\pi)}$.

# B Formal statement of bounds on the minimal information ratio

In Section 4, we listed several bounds on the minimal information ratio, but only provided a formal statement of the result on linear bandit problems. Here we provide a more complete description of three other bounds. The proofs are given in Appendix D.

## B.1 Worst case bound

The next proposition shows that $\Psi_t^*$ is never larger than $|\mathcal{A}|/2$. That is, there is always an action sampling distribution $\pi$ such that $\Delta_t(\pi)^2 \leq (|\mathcal{A}|/2)g_t(\pi)$. In the next section, we will will show that under different information structures the ratio between regret and information gain can be much smaller, which leads to stronger theoretical guarantees.

**Proposition 4.** *For any* $t \in \mathbb{N}$, $\Psi_t^* \leq \leq |\mathcal{A}|/2$ *almost surely.*

Combining Proposition 4 with Proposition 2 shows that $\mathbb{E}\left[ \mathrm{Regret}\left( \pi^{\mathrm{IDS}}, T \right) \right] \leq \sqrt{\frac{1}{2} |\mathcal{A}| H(\alpha_1) T}$.

## B.2 Full information

Our focus in this paper is on problems with *partial feedback*. For such problems, what the decision maker observes depends on the actions selected, which leads to a tension between exploration and exploitation. Problems with full information arise as an extreme point of our formulation where the outcome $Y_t(a)$ is perfectly revealed by observing $Y_t(\tilde{a})$ for some $\tilde{a} \neq a$; what is learned does not depend on the selected action. The next proposition shows that under full information, the minimal information ratio is bounded by $1/2$.

**Proposition 5.** *Suppose for each* $t \in \mathbb{N}$ *there is a random variable* $Z_t : \Omega \to \mathcal{Z}$ *such that for each* $a \in \mathcal{A}$, $Y_t(a) = (a, Z_t)$. *Then for all* $t \in \mathbb{N}$, $\Psi_t^* \leq \frac{1}{2}$ *almost surely.*

Combining this result with Proposition 2 shows $\mathbb{E}\left[ \mathrm{Regret}(T, \pi^{\mathrm{IDS}}) \right] \leq \sqrt{\frac{1}{2} H(\alpha_1) T}$. Further, a worst–case bound on the entropy of $\alpha_1$ shows that $\mathbb{E}\left[ \mathrm{Regret}(T, \pi^{\mathrm{IDS}}) \right] \leq \sqrt{\frac{1}{2} \log(|\mathcal{A}|) T}$. Dani

et al. [12] show this bound is order optimal, in the sense that for any time horizon $T$ and number of actions $|\mathcal{A}|$ there exists a prior distribution over $p^*$ under which $\inf_\pi \mathbb{E}\left[\text{Regret}(T, \pi)\right] \geq c_0\sqrt{\log(|\mathcal{A}|)T}$ where $c_0$ is a numerical constant that does not depend on $|\mathcal{A}|$ or $T$. The bound here improves upon this worst case bound since $H(\alpha_1)$ can be much smaller than $\log(|\mathcal{A}|)$ when the prior distribution is informative.

### B.3 Combinatorial action sets and "semi–bandit" feedback

To motivate the information structure studied here, consider a simple resource allocation problem. There are $d$ possible projects, but the decision–maker can allocate resources to at most $m \leq d$ of them at a time. At time $t$, project $i \in \{1, .., d\}$ yields a random reward $\theta_{t,i}$, and the reward from selecting a subset of projects $a \in \mathcal{A} \subset \{a' \subset \{0, 1, ..., d\} : |a'| \leq m\}$ is $m^{-1}\sum_{i \in \mathcal{A}} \theta_{t,i}$. In the linear bandit formulation of this problem, upon choosing a subset of projects $a$ the agent would only observe the overall reward $m^{-1}\sum_{i \in a} \theta_{t,i}$. It may be natural instead to assume that the outcome of each selected project $(\theta_{t,i} : i \in a)$ is observed. This type of observation structure is sometimes called "semi–bandit" feedback [3].

A naive application of Proposition 3 to address this problem would show $\Psi_t^* \leq d/2$. The next proposition shows that since the entire parameter vector $(\theta_{t,i} : i \in a)$ is observed upon selecting action $a$, we can provide an improved bound on the information ratio.

**Proposition 6.** *Suppose $\mathcal{A} \subset \{a \subset \{0, 1, ..., d\} : |a| \leq m\}$, and that there are random variables $(\theta_{t,i} : t \in \mathbb{N}, i \in \{1, ..., d\})$ such that*

$$Y_t(a) = (\theta_{t,i} : i \in a) \quad and \quad R\left(Y_t(a)\right) = \frac{1}{m}\sum_{i \in a} \theta_{t,i}.$$

*Assume that the random variables $\{\theta_{t,i} : i \in \{1, ..., d\}\}$ are independent conditioned on $\mathcal{F}_{t-1}$ and $\theta_{t,i} \in [\frac{-1}{2}, \frac{1}{2}]$ almost surely for each $(t, i)$. Then for all $t \in \mathbb{N}$, $\Psi_t^* \leq \frac{d}{2m^2}$ almost surely.*

In this problem, there are as many as $\binom{d}{m}$ actions, but because information-directed sampling exploits the structure relating actions to one another, its regret is only polynomial in $m$ and $d$. In particular, combining Proposition 6 with Proposition 2 shows $\mathbb{E}\left[\text{Regret}(T, \pi^{\text{IDS}})\right] \leq \frac{1}{m}\sqrt{\frac{d}{2}H(\alpha_1)T}$. Since $H(\alpha_1) \leq \log|\mathcal{A}| = O(m\log(\frac{d}{m}))$ this also yields a bound of order $\sqrt{\frac{d}{m}\log\left(\frac{d}{m}\right)T}$. As shown by Audibert et al. [3], the lower bound[6] for this problem is of order $\sqrt{\frac{d}{m}T}$, so our bound is order optimal up to a $\sqrt{\log(\frac{d}{m})}$ factor.

## C Algorithms used in numerical experiments

### C.1 Optimization

Algorithm 1 uses Proposition 1 to provide a procedure for choosing an action. For a problem with $|\mathcal{A}| = K$ actions, the algorithm requires inputs $\Delta \in \mathbb{R}_+^K$ and $g \in \mathbb{R}_+^K$ specifying respectively the expected regret and information gain of each action. The distribution that minimizes (5) is computed by looping over all pairs of actions $(i, j) \in \mathcal{A} \times \mathcal{A}$ and finding the optimal probability of playing $i$ instead of $j$. Finding this probability is particularly efficient because the objective function is convex. Golden section search, for example, provides a very efficient

**Algorithm 1** chooseAction($\Delta \in \mathbb{R}_+^K, g \in \mathbb{R}_+^K$)

---

1: **Initialize** opt $\leftarrow \infty$
2: **Calculate Optimal Sampling Distribution**
3: **for** $j \in \{1, .., K-1\}$ **do**
4:     **for** $i \in \{i+1, .., K\}$ **do**
5:         $q \leftarrow \arg\min_{q' \in [0,1]} [q'\Delta_i + (1-q)\Delta_j]^2 / [q'g_i + (1-q')g_j]$
6:         objectiveValue $\leftarrow [q\Delta_i + (1-q)\Delta_j]^2 / [qg_i + (1-q)g_j]$
7:         **if** objectiveValue $<$ opt **then**
8:             $(i^*, j^*, q^*) \leftarrow (i, j, q)$
9:             opt $\leftarrow$ objectiveValue
10:         **end if**
11:     **end for**
12: **end for**
13:
14: **Select Action**:
15: Sample $U \sim \text{Uniform}([0,1])$
16: **if** $U < q^*$ **then**
17:     Play $i^*$
18: **else**
19:     Play $j^*$
20: **end if**

---

method for optimizing a convex function over $[0, 1]$. In addition, in this case, any solution to $\frac{d}{dq'}[q'\Delta_i + (1-q)\Delta_j]^2 / [q'g_i + (1-q')g_j] = 0$ is given by the solution to a quadratic equation, and therefore can be expressed in closed form.

### C.2 Beta–Bernoulli

Here we will present an implementation of information-directed sampling for the Beta–Bernoulli experiment described in Section 6. Consider a multi-armed bandit problem with binary rewards and $K$ independent arms denoted by $\mathcal{A} = \{a_1, ..., a_K\}$. The mean reward $X_i$ of each arm $a_i$ is drawn from a Beta prior distribution, and the mean of separate arms are modeled independently.

Because the Beta distribution is a conjugate prior for the Bernoulli distribution, the posterior distribution of each $X_i$ is a Beta distribution. The parameters $(\beta_i^1, \beta_i^2)$ of this distribution can be updated easily. Let $f_i(x)$ and $F_i(x)$ denote respectively the PDF and CDF of the posterior distribution $X_i$. The posterior probability that $A^* = a_i$ can be written as

$$\mathbb{P}\left(\bigcap_{j \neq i} \{X_j \leq X_i\}\right) = \int_0^1 f_i(x) \mathbb{P}\left(\bigcap_{j \neq i} \{X_j \leq x\} \bigg| X_i = x\right) dx = \int_0^1 f_i(x) \left(\prod_{j \neq i} F_j(x)\right) dx$$

$$= \int_0^1 \left[\frac{f_i(x)}{F_i(x)}\right] \overline{F}(x) dx$$

where $\overline{F} : x \mapsto \prod_{i=1}^K F_i(x)$.

Algorithm 2 uses this expression to compute the posterior probability $\alpha_i$ that an action $a_i$ is optimal. To compute the information gain $g_j$ of action $j$, we use Fact **??**. Let $M_{i,j} := \mathbb{E}[X_j | X_k \leq X_i \; \forall k]$ denote the expected value of $X_j$ given that action $i$ is optimal. Step 18 computes the information gain $g_j$ of action $a_j$ as the expected Kullback Leibler divergence between a Bernoulli distribution with mean $M_{i,j}$ and the posterior distribution at action $j$, which is Bernoulli with parameter $\beta_j^1 / (\beta_j^1 + \beta_j^2)$.

---
**Algorithm 2** Beta-Bernoulli IDS
---

1: **Initialize**: Input posterior parameters:
   $(\beta^1 \in \mathbb{R}^K, \beta^2 \in R^K)$
2: $f_i(x) := \text{Beta.pdf}(x|\beta_i^1, \beta_i^2)$ for $i \in \{1, .., K\}$
3: $F_i(x) := \text{Beta.cdf}(x|\beta_i^1, \beta_i^2)$ for $i \in \{1, .., K\}$
4: $\overline{F}(x) := \prod_{i=1}^K F_i(x)$
5: $Q_i(x) := \int_0^x y f_i(y) dy$ for $i \in \{1, .., K\}$
6: $\text{KL}(p_1||p_2) := p_1 \log\left(\frac{p_1}{p_2}\right) + (1 - p_1)\log\left(\frac{1-p_1}{1-p_2}\right)$
7:
8: **Calculate optimal action probabilities**:
9: **for** $i \in \{1, .., K\}$ **do**
10:      $\alpha_i \leftarrow \int_0^1 \left[\frac{f_i(x)}{F_i(x)}\right] \overline{F}(x) dx$
11: **end for**
12:
13: **Calculate Information Gain**
14: **for** $(i, j) \in \{1, .., K\} \times \{1, .., K\}$ **do**
15:      **if** $(i == j)$ **then**
16:           $M_{i,i} \leftarrow \frac{1}{\alpha_i} \int_0^1 \left[\frac{x f_i(x)}{F_i(x)}\right] \overline{F}(x) dx$
17:      **else**
18:           $M_{i,j} \leftarrow \frac{1}{\alpha_i} \int_0^1 \left[\frac{f_i(x)\overline{F}(x)}{F_i(x)F_j(x)}\right] Q_j(x) dx$
19:      **end if**
20: **end for**
21:
22: **Fill in problem data**
23: $\rho^* \leftarrow \sum_{i=1}^K \alpha_i M_{i,i}$
24: **for** $i \in \{1, ..., K\}$ **do**
25:      $\Delta_i \leftarrow \rho^* - \frac{\beta_i^1}{\beta_i^1 + \beta_i^2}$ for $i \in \{1, .., K\}$
26:      $g_i \leftarrow \sum_{j=1}^K \alpha_j \text{KL}\left(M_{j,i} || \frac{\beta_i^1}{\beta_i^1 + \beta_i^2}\right)$
27: **end for**
28:
29: $\text{chooseAction}(\Delta, g)$
---

Finally, the algorithm computes the expected reward of the optimal action $\rho^* = \mathbb{E}\left[\max_j X_j\right]$ and uses that to compute the expected regret of action $j$:

$$\Delta_i = \mathbb{E}\left[\max_j X_j - X_i\right] = \rho^* - \frac{\beta_i^1}{(\beta_i^1 + \beta_i^2)}.$$

Practical implementations of this algorithm can approximate each definite integral by evaluating the integrand at a discrete grid of points in $\{x^1, ..., x^n\} \subset [0, 1]$. The values of $f_i(x), F_i(x), Q_i(x)$ and $\overline{F}(x)$ can be computed and stored for each value of $x$ in this grid. In each period, the posterior distribution of only a single action is updated, and hence these values need to be updated for only one action each period.

Steps 14-21 are the most computationally intensive part of the algorithm. The computational cost of these steps scales as $K^2 n$ where $K$ is the number of actions and $n$ is the number of points used in the discretization of [0,1].

### C.3 Linear mean-based information-directed sampling

This section provides an implementation of mean-based information-directed sampling for the problem of linear optimization under bandit feedback. Consider a problem where the action set $\mathcal{A}$ is a fi-

nite subset of $\mathbb{R}^d$, and whenever an action $a$ is sampled, only the resulting reward $Y_t(a) = R(Y_t(a))$ is observed. There is an unknown parameter $\theta^* \in \mathbb{R}^d$ such that for each $a \in \mathcal{A}$ the expected reward of $a$ is $a^T \theta^*$.

In Subsection 6 we introduced the term

$$g_t^{\mathrm{ME}}(a) = \mathop{\mathbb{E}}_{a^* \sim \alpha_t} \left[ D_{\mathrm{ME}} \left( p_{t,a}(\cdot | a^*) \,\|\, p_{t,a} \right)^2 \right] \tag{8}$$

where

$$
\begin{aligned}
D_{\mathrm{ME}} \left( p_{t,a}(\cdot | a^*) \,\|\, p_{t,a} \right)^2 &:= \left( \mathop{\mathbb{E}}_{y \sim p_{t,a}(\cdot | a^*)} [R(y)] - \mathop{\mathbb{E}}_{y \sim p_{t,a}} [R(y)] \right)^2 \\
&= \left( \mathbb{E} \left[ R(Y_t(a)) | \mathcal{F}_{t-1}, A^* = a^* \right] - \mathbb{E} \left[ R(Y_t(a)) | \mathcal{F}_{t-1} \right] \right)^2.
\end{aligned}
$$

We will show that for this problem, $g_t^{\mathrm{ME}}(a)$ takes on a particularly simple form, and will present an algorithm that leverages this. Write $\mu_t = \mathbb{E} \left[ \theta^* | \mathcal{F}_{t-1} \right]$ and write $\mu_t^{(a^*)} = \mathbb{E} \left[ \theta^* | \mathcal{F}_{t-1}, A^* = a^* \right]$. Define

$$\mathrm{Cov}_t(X) = \mathbb{E} \left[ \left( X - \mathbb{E} \left[ X | \mathcal{F}_{t-1} \right] \right) \left( X - \mathbb{E} \left[ X | \mathcal{F}_{t-1} \right] \right)^T | \mathcal{F}_{t-1} \right]$$

to be the posterior covariance of a random variable $X : \Omega \to \mathbb{R}^d$. Then,

$$D_{\mathrm{ME}} \left( p_{t,a}(\cdot | a^*) \,\|\, p_{t,a} \right)^2 = a^T \left( [\mu_t^{(a^*)} - \mu_t][\mu_t^{(a^*)} - \mu_t]^T \right) a$$

and therefore

$$g_t^{\mathrm{ME}}(a) = a^T L_t a$$

where

$$L_t = \mathop{\mathbb{E}}_{a^* \sim \alpha_t} [\mu_t^{(a^*)} - \mu_t][\mu_t^{(a^*)} - \mu_t]^T = \mathrm{Cov}_t \left( \mu_t^{(A^*)} \right). \tag{9}$$

is exactly the posterior covariance matrix of $\mu_t^{(A^*)}$.

Algorithm 3 presents a simulation based procedure that computes $g_t^{\mathrm{ME}}(a)$ and $\Delta_t(a)$ and selects an action according to the distribution $\pi_t^{\mathrm{IDS_{ME}}}$. This algorithm requires the ability to generate a large number of samples, denoted by $M \in \mathbb{N}$ in the algorithm, from the posterior distribution of $\theta^*$, which is denoted by $P(\cdot)$ in the algorithm. The actions set $\mathcal{A} = \{a_1, ..., a_K\}$ is represented by a matrix $A \in \mathbb{R}^{K \times d}$ where the $i$th row of $A$ is the action feature vector $a_i \in \mathbb{R}^d$. The algorithm directly approximates the matrix $L_t$ that appears in equation (9). It does this by sampling parameters from the posterior distribution of $\theta^*$, and, for each action $a$, tracking the number of times $a$ was optimal and the sample average of parameters under which $a$ was optimal. From these samples, it can also compute an estimated vector $R \in \mathbb{R}^K$ of the mean reward from each action and an estimate $p^* \in \mathbb{R}$ of the expected reward from the optimal action $A^*$.

# D    Proofs

## D.1    Proof of Proposition 1

*Proof.* First, we show the function $\Psi : \pi \mapsto \left( \pi^T \Delta \right)^2 / \pi^T g$ is convex on $\{ \pi \in \mathbb{R}^K | \pi^T g > 0 \}$. As shown in Chapter 3 of Boyd and Vandenberghe [5], $f : (x, y) \mapsto x^2/y$ is convex over $\{ (x, y) \in \mathbb{R}^2 : y > 0 \}$. The function $h : \pi \mapsto (\pi^T \Delta, \pi^T g) \in \mathbb{R}^2$ is affine. Since convexity is preserved under composition with an affine function, the function $\Psi = g \circ h$ is convex.

We now consider the second claim. Let $\Psi^* \in \mathbb{R}$ denote the optimal objective value for the minimization problem (6). Define the function

$$\rho(\pi) = \left( \pi^T \Delta \right)^2 - \Psi^* \left( \pi^T g \right)$$

and consider minimizing $\rho(\pi)$ over all probability vectors $\pi \in \{ v \in \mathbb{R}^K : v^T e = 1, v \geq 0 \}$. Since $\Psi^*$ is the minimal value of (6), for any feasible $\pi$, $\rho(\pi) \geq 0$, but for $\pi^*$ minimizing (6), $\rho(\pi^*) = 0$. Therefore the set of minimizers of $\rho(\cdot)$ is the same as the set of minimizers of (6). We will now show that there is a minimizer of $\rho(\pi)$ with at most two nonzero components.

---
**Algorithm 3** Linear Information-Directed Sampling
---
1: **Initialize**: Input $A \in \mathbb{R}^{K \times d}$, $M \in \mathbb{N}$ and posterior distribution $P(\theta)$.
2: **for** $i \in \{1, .., K\}$ **do**
3:     $s^{(i)} \leftarrow 0 \in \mathbb{R}^d$
4:     $n_i \leftarrow 0 \in \mathbb{R}$
5: **end for**
6:
7: **Perform Monte Carlo**:
8: **for** $m \in \{1, .., M\}$ **do**
9:     Sample $\theta \sim P(\cdot)$
10:     $I \leftarrow \arg\max_i\{(A\theta)_i\}$
11:     $n_I \mathrel{+}= 1$
12:     $s^{(I)} \mathrel{+}= \theta$
13: **end for**
14:
15: **Calculate Problem Data From Monte Carlo Totals**
16: $\mu \leftarrow \frac{1}{M} \sum_{i=1}^{K} s^{(i)}$
17: $R \leftarrow A\mu \in \mathbb{R}^K$
18: **for** $i \in \{1, .., K\}$ **do**
19:     $\mu^{(i)} \leftarrow s^{(i)}/n_i$
20:     $\alpha_i \leftarrow n_i/M$
21: **end for**
22: $L \leftarrow \sum_{i=1}^{K} \alpha_i \left(\mu^{(i)} - \mu\right)\left(\mu^{(i)} - \mu\right)^T \in \mathbb{R}^{d \times d}$
23: $\rho^* \leftarrow \sum_{i=1}^{K} \alpha_i [a_i^T \mu^{(i)}] \in \mathbb{R}$
24: **for** $i \in \{1, .., K\}$ **do**
25:     $g_i \leftarrow a_i^T L a_i \in \mathbb{R}$
26:     $\Delta_i \leftarrow \rho^* - a_i^T \mu \in \mathbb{R}$
27: **end for**
28:
29: chooseAction$(\Delta, g)$
---

Fix a minimizer $\pi^*$ of $\rho(\cdot)$. Differentiating of $\rho(\pi)$ with respect to $\pi$ at $\pi = \pi^*$ yields

$$\frac{\partial}{\partial \pi}\rho(\pi^*) = 2\left(\Delta^T \pi^*\right)\Delta - \Psi^* g$$
$$= 2L^*\Delta - \Psi^* g$$

where $L^* = \Delta^T \pi^*$ is the expected instantaneous regret of the sampling distribution $\pi^*$. Let $d^* = \min_i \frac{\partial}{\partial \pi_i}\rho(\pi^*)$ denote the smallest partial derivative of $\rho$ at $\pi^*$. It must be the case that any $i$ with $\pi_i^* > 0$ satisfies $d^* = \frac{\partial}{\partial \pi_i}\rho(\pi^*)$, as otherwise transferring probability from action $a_i$ could lead to strictly lower cost. This shows that for each $i$ with $\pi_i^* > 0$,

$$g_i = \frac{-d^*}{\Psi^*} + \frac{2L^*}{\Psi^*}\Delta_i. \tag{10}$$

Let $i_1, .., i_m$ be the indices such that $\pi_{i_k}^* > 0$ and $g_{i_1} \geq g_{i_2} \geq ... \geq g_{i_m}$. Then we can choose a $\beta \in [0, 1]$ so that

$$\sum_{k=1}^m \pi_{i_k}^* g_{i_k} = \beta g_{i_1} + (1 - \beta)g_{i_m}.$$

By equation (10), this implies as well that $\sum_{k=1}^m \pi_{i_k}^* \Delta_{i_k} = \beta \Delta_{i_1} + (1 - \beta)\Delta_{i_m}$, and hence that the sampling distribution that plays $a_{i_1}$ with probability $\beta$ and $a_{i_m}$ otherwise has the same instantaneous expected regret and the same expected information gain as $\pi^*$. That is, starting with a general sampling distribution $\pi^*$ that maximizes $\rho(\pi)$, we showed there is a sampling distribution with support over at most two actions attains the same objective value and hence that also maximizes $\rho(\pi)$. $\qquad\square$

## D.2 Proof of Proposition 2

The following fact expresses the mutual information between $A^*$ and $Y_t(a)$ as the as the expected reduction in entropy due to observing $Y_t(a)$.

**Fact 1.** *(Lemma 5.5.6 of Gray [16])*

$$I_t\left(A^*; Y_t(a)\right) = \mathbb{E}\left[H(\alpha_t) - H(\alpha_{t+1})|A_t = a, \mathcal{F}_{t-1}\right]$$

We now prove Proposition 2,

*Proof.* By definition, if $\Psi_t(\pi_t) \leq \lambda$, then $\Delta_t(\pi_t) \leq \sqrt{\lambda}\sqrt{g_t(\pi_t)}$. Therefore,

$$\mathbb{E}\left[\mathrm{Regret}(T, \pi)\right] = \mathbb{E}\sum_{t=1}^T \Delta_t(\pi_t) \leq \sqrt{\lambda}\mathbb{E}\sum_{t=1}^T \sqrt{g_t(\pi_t)} \overset{(a)}{\leq} \sqrt{\lambda T}\sqrt{\mathbb{E}\sum_{t=1}^T g_t(\pi_t)} \overset{(b)}{\leq} \sqrt{\lambda H(\alpha_1) T}.$$

Inequality (a) follows from Hölder's inequality. To show inequality (b), note that if actions are selected according to a policy $\pi = (\pi_1, \pi_2, ...)$, then

$$\mathbb{E}\sum_{t=1}^T g_t(\pi_t) = \mathbb{E}\sum_{t=1}^T \mathbb{E}\left[H(\alpha_t - H(\alpha_{t+1})|\mathcal{F}_{t-1}\right] = \mathbb{E}\sum_{t=1}^T \left(H(\alpha_t - H(\alpha_{t+1}))\right) = H(\alpha_1) - H(\alpha_{T+1}) \leq H(\alpha_1),$$

where the first equality relies on Fact 1 and the tower property of conditional expectation and the final inequality follows from the non-negativity of entropy. $\qquad\square$

## D.3 Proof of bounds on the information ratio

Here we leverage the tools of a very recent analysis of Thompson sampling [25] to provide bounds on the information ratio of IDS. Since $\Psi_t^* = \min_\pi \Psi_t(\pi) \leq \Psi_t(\pi_t^{\mathrm{TS}})$, the bounds on $(\pi_t^{\mathrm{TS}})$ provided by Russo and Van Roy [25] immediately yield bounds on the minimal information ratio.

### D.3.1 Proof of Proposition 4

*Proof.* Proposition 3 of Russo and Van Roy [25], shows

$$\Psi_t(\pi_t^{\mathrm{TS}}) \leq \frac{|\mathcal{A}|}{2}$$

almost surely for any $t \in \mathbb{N}$. Note that the term $\Gamma_t$ in that paper is exactly $\Psi_t(\pi_t^{\mathrm{TS}})^2$. The result then follows since

$$\Psi_t^* \stackrel{\mathrm{Def}}{=} \min_{\pi \in \mathcal{D}(\mathcal{A})} \Psi_t(\pi) \leq \Psi_t(\pi_t^{\mathrm{TS}}).$$

$\square$

### D.3.2 Proof of Proposition 5

*Proof.* See Proposition 4 in [25], which shows $\Psi_t(\pi_t^{\mathrm{TS}}) \leq 1/2$.

$\square$

### D.3.3 Proof of Proposition 3

*Proof.* See Proposition 5 of Russo and Van Roy [25], which shows $\Psi_t(\pi_t^{\mathrm{TS}}) \leq d/2$.

$\square$

### D.3.4 Proof of Proposition 6

Proposition 6 of Russo and Van Roy [25], shows

$$\Psi_t(\pi_t^{\mathrm{TS}}) \leq \frac{d}{2m^2}$$

almost surely for any $t \in \mathbb{N}$. Note that the term $\Gamma_t$ in that paper is exactly $\Psi_t(\pi_t^{\mathrm{TS}})^2$. The result then follows since

$$\Psi_t^* \stackrel{\mathrm{Def}}{=} \min_{\pi \in \mathcal{D}(\mathcal{A})} \Psi_t(\pi) \leq \Psi_t(\pi_t^{\mathrm{TS}}).$$

## Footnotes

[5]For details on the derivation of this fact when $Y_t(a)$ is a general random variable, see the appendix of [25].

[6] In their formulation, the reward from selecting action $a$ is $\sum_{i \in a} \theta_{t,i}$, which is $m$ times larger than in our formulation. The lower bound stated in their paper is therefore of order $\sqrt{mdT}$. They don't provide a complete proof of their result, but note that it follows from standard lower bounds in the bandit literature. In the proof of Theorem 5 in that paper, they construct an example in which the decision maker plays $m$ bandit games in parallel, each with $d/m$ actions. Using that example, and the standard bandit lower bound (see Theorem 3.5 of Bubeck and Cesa-Bianchi [7]), the agent's regret from each component must be at least $\sqrt{\frac{d}{m}T}$, and hence her overall expected regret is lower bounded by a term of order $m\sqrt{\frac{d}{m}T} = \sqrt{mdT}$.