[Reviews · NeurIPS 2014]

Submitted by Assigned_Reviewer_4

The paper presents Information-Directed Sampling (IDS), a novel algorithm for multi-armed bandit problems with structural properties. Under this algorithm, an action is sampled to minimize the ratio of the square of the expected single-period regret and a measure of information gain. The authors present a generic regret analysis of IDS and exemplify the regret upper bound through simple examples. The authors further perform numerical evaluations of IDS and show that it outperforms most of the state-of-the-art algorithms proposed so far on the examples of MAB with independent arms and of linear bandit problems.

The paper is easy to read, and introduce a novel idea in bandit optimization. The way exploration and exploitation is balanced is new and intuitively sound. There are several points in the paper whose clarity could be improved.
1. About the complexity of solving optimization (5), how does it depend on the underlying known structure? The authors exemplify their algorithm for independent arms and for linear bandits, do they have an efficient implementation for more general structure?
2. The bounds derived in section 4 are interesting, but concerns worse cases. When the prior on the distribution of the reward distributions concentrates on a single distribution, would it be possible to derive more precise regret bounds (problem specific bounds)? In particular, for the case of independent arms, it should be interesting to state a problem-specific regret upper bound under IDS. When designing a new bandit algorithm, I think the minimum requirement would be its asymptotical optimality in case of independent arms.
3. In Section 6, the authors do not experiment with KL-UCB, known to outperform UCB for the independent arm case. It seems crucial to compare the performance of IDS with that of KL-UCB.

When the problem exhibits a strong structure, it is in general hard to compute the posterior distribution of the parameters given the observation. In particular, this problem is already limiting when implementing the Thomson sampling algorithm. It seems even harder to implement IDS in this case. I think it would be worth commenting this issue, and explain how to circumvent it.
Summary: Overall, the paper presents a novel idea for generic MAB problems, and the proposed algorithm does seem to perform very well. The paper is technically strong, and novel.

Submitted by Assigned_Reviewer_9

This paper presents a new general algorithm for Bayesian bandit problems, based on a tradeoff between the "information gain" of an action and its contribution to the regret. The idea is new and interesting, and the algorithm appears to display good performance on numerical simulations. However the general results stated for the method are comparable to that of other approaches and the claim that it can "dramatically outperform upper confidence bound algorithms and Thompson sampling" (in the introduction) appear to be excessive in light of the elements provided in Section 5 and 6. In addition, details of the algorithms given in the supplementary material suggest that the implementation is much more involved than that of UCB or Thompson Sampling.

Regarding the latter aspect it is difficult to understand by reading the paper to which extend the algorithm can really be implemented and will be efficient. In particular, for which bandit problems are the quantities \alpha_t, g_t, \Delta_t easy to compute? The implementation for Beta-Bernoulli models can only be found in the supplementary material, and the authors almost never mention this classical example of bandit model in the main text. Explanations on how to compute g_t and \Delta_t should be given as early in the text as possible (and certainly not in the supplementary material only).

The intuition on why the information gain is defined this way is not obvious: there are plenty of ways of realizing a tradoff between the two quantities \Delta_t and g_t: why not minimizing \Delta_t + 1/g_t for example? The proof of Proposition 2 should be provided in the main text, as (1) it is the theoretical cornerstone of the analysis (the bounds on \Phi_t(\pi^TS) following from previous work if I get it right); (2) it helps understanding why \Pi_t is chosen this way (or at least why this choice make the proof work!); (3) it is short.

The value of the counter-examples in Section 5 is somewhat uncertain as they appear to rely on very specific assumptions. In particular, the first counter-example relates to the very artificial setting of sparse linear bandits without observation noise and known 1-sparsity. In light of the authors' answer in the rebuttal regarding this comment, it is clear that the argument also relies on the fact that the arms are L1-normalized vectors of {0,1}^d, as omitting the normalization would make Thompson sampling select arms with more than one non-zero coordinate (just as the proposed method does). Note that in this (artificial) setting, L1-normalizing the arms is a curious choice as it makes all the arms that have more than one non-zero component inadmissible in the linear bandit model as they are convex combinations of others (ie. they could be removed from start, leaving only the d arms that have a single non-zero component). This being said, the main question here is what can be learned more generally from these very specific situations?

In the numerical experiments, please specify that Figure (a) displays the Bayesian regret (drawing problems from the priors). You should also comment more on the difficulty of implementing IDS in the Bernoulli case compared to the other approaches. There is also big typo in the Lai and Robbins lower bound: the sum is before the fraction! (line 394). I didn't check the complexity of implementation for linear bandits: could you comment on how much complex it is compared to Thompson Sampling? Could you also give precisions on the modified version on GP-UCB that is mentioned in the text?

In Section 1, it may be worth mentioning that the use of the mutual information to select the most informative actions is also a largely used principle in sensor networks (googling "information-based sensor" of "information-theoretic sensor" will bring up plenty of related references), although the combination with a regret measure is quite probably novel.

Minor typos: 124: It's -> It is; 150: $R_t$ should be $R$ I guess; 177: "_the_ design"; 273: it should be \cP_t (the subscript t is missing); 562: 988888 ???.
Summary: Interesting idea for defining new Bayesian strategies for reward maximization. The paper is however not fully conclusive regarding the merits and the applicability of the approach.

Submitted by Assigned_Reviewer_29

In this paper the authors present a novel algorithm for solving the explore-exploit dilemma in online optimization. The algorithm, information directed sampling, selects actions in a way that tries to minimize regret and maximize information about the most informative option. These two desires are balanced in their objective function, the information ratio, that is the ratio of the square expected regret to the information about the optimal action. This algorithm is applicable in a wide variety of situations and they show it has excellent theoretical properties – achieving logarithmic regret for simple bandit problems and sublinear regret in cases in which UCB and Thompson sampling perform poorly.

This is an impressive piece of work and will no doubt be of interest to attendees at NIPS. Like most papers in the explore-exploit world, however, this paper is not very well written. In my opinion it is overly dense and makes very little effort to be accessible to people outside of the field. This is a shame, because the results are really nice and I believe it wouldn’t take much to make it easier to read.

Concrete suggestions for doing this:

1) Where does the information ratio come from? Can you give us any intuition as to why we need regret squared / information? Certainly it makes sense to trade of information and reward, but why trade them off as a ratio and not e.g. subtract them? Why regret squared?

2) For the casual reader, I think the paragraph on Randomized policies is unnecessary. This could be moved to the supplement and a simple sentence mentioning that action selection is stochastic – where the exact type of stochasticity is described in the appendix.

3) Is there an intuition for the mean-based IDS? Also can you give any sense of the efficiency savings with this algorithm? For example for the examples in Section 6.

In summary a decent paper – should probably be accepted – just wish it was more accessible.

Typos:
“also only considers only the” -> two onlys

Supplement pg 12 “Fact ??.”

UPDATE BASED ON REVIEWER DISCUSSION:
One of the reviewers makes a good point about the lack of detail on the implementation of the algorithm. In light of this I am dropping down to a 7.
Summary: A new explore-exploit algorithm that outperforms the current state of the art in a variety of interesting tasks. Good paper but could be more accessible.
Author Feedback
Author rebuttal: We thank the reviewers for their thoughtful feedback.

Allow us to offer our perspective on this project: We consider an approach to exploration that is conceptually very different than most popular approaches. It is based on carefully quantifying the information actions provide about the optimum, instead of the dominant 'optimism under uncertainty' paradigm. Our aim is not to provide a specific numerical routine for a specific application. Instead, we view IDS as an abstract concept that may lead to useful insights and/or numerical methods for many classes of problems. We have provided efficient numerical methods based on IDS for a few problem classes, including Bernoulli, Gaussian, and linear bandit problems. In the long run, we think this line of thought could lead to much broader practical benefits. We'll work to clarify this perspective in the introduction.

We've been thinking about this direction for a couple of years now, and tried to share everything we've learned about IDS. We provided (1) several general theoretical guarantees, (2) simple examples that can't be addressed adequately by popular approaches, (3) simulation experiments for some widely studied bandit problems, and (4) numerical procedures for implementing IDS in a few specific settings. We feel that each of these should be valuable to the NIPS community.

In retrospect, however, that may be too much material for an 8-page paper, and it has affected the presentation. We hope to revise the paper to provide more high level intuition, to clarify our perspective on the algorithm, and to provide a discussion of open questions and the limitations of our work. We're particularly concerned by the first reviewer's comment that the paper is inaccessible. We will try to address this, and thank the reviewer for his/her suggestions.

----------------------------
*Why does IDS take this form*

In short, this is one way of balancing regret and information gain, but it's not the only effective way, and we can't claim at this point that it's *the* "right" way. The particular form of IDS allows us to establish strong and elegant regret bounds, and is effective in experiments. The ratio form also has a nice intuition behind it: one minimizes a measure of the cost-per-bit of information acquired.

--------------------
*Computation of IDS*

There are many open questions regarding the computation of IDS. We cut a discussion of this from the conclusion due to space constraints. In light of the reviewers' comments, we will include this section.

As stated in the introduction, "IDS solves a single-period optimization problem as a proxy to an intractable multi-period problem. However, solution of this single-period problem can itself be computationally demanding, especially in cases where the number of actions is enormous or mutual information is difficult to evaluate."

Again, we view IDS as an abstract procedure, and a conceptual target. To carry out experiments we've developed efficient numerical methods for the specific yet relevant and widely studied cases of Bernoulli, Gaussian, and linear bandits. But we don't know of an efficient method of computing the mutual information terms used by the algorithm that applies across *all* online optimization problems. In the worst-case, time-consuming Monte-Carlo simulations may be required to evaluate these terms.

While this is limiting, we feel the developments presented here could generate significant interest among the NIPS audience. And ultimately, "we hope that our development and analysis of IDS facilitate the future design of efficient algorithms that capture its benefits."

-------------------------------
*Sparse linear bandit example*

We disagree with the comments of reviewer 3 on this example. First, we are certain that the main claims are correct, and apply to forms of TS and UCB that take the linear structure into account. As you state, TS will sample a parameter with a single nonzero component and play the action with maximal dot product with this parameter vector. But the action with the maximal dot product also has only one positive component: the same one as the sampled parameter. This is essentially our claim in lines 293-295 that A^*=\theta^*.

It's true that the "counter-examples" are artificially simple. This is by design. Ideally, a counter-example is transparent and is suggestive of challenges that one expects to arise in more complicated settings. In addition, simpler counter-examples provide stronger results. We show that TS and UCB cannot learn in time sublinear in the dimension even in the easiest case of a sparse-linear bandit where (1) the problem is 1-sparse and (2) there is no observation noise. Adding observation noise, for example, would not make the problem any easier for TS or UCB.

We feel this is a nontrivial contribution of this work. We've talked with several researchers who were interested in applying TS to sparse problems, and were unaware of its shortcomings.

-------------------
*Worst case bounds*

When the prior concentrates on a single action being optimal, the entropy of the optimal-action distribution tends to zero and so do our regret bounds.

We may not have entirely understood the reviewer's comment about only proving worst-case bounds. Perhaps you're referring to the fact that our bounds are "gap-independent" and hence scale with \sqrt{T} instead of log(T). Which scaling is more appropriate depends on whether the horizon is large compared to the gaps between the reward distributions. We feel that the \sqrt{T} scaling is more appropriate for many online optimization problems, like linear bandit problems, for example, where some arms are nearly indistinguishable from others.

At this point, we don’t have a complete proof that IDS attains the lower bound of Lai and Robbins, but have worked on this.

---------
*KL UCB*

If accepted, we will include KL-UCB in the numerical experiments section.